# Characterization of Phage vB_SalM_SPJ41 and the Reduction of Risk of Antibiotic-Resistant *Salmonella enterica* Contamination in Two Ready-to-Eat Foods

**DOI:** 10.3390/antibiotics12020364

**Published:** 2023-02-09

**Authors:** Tengteng Li, Hong Chen, Jiayi Zhao, Zhenxiang Tao, Weiqing Lan, Yong Zhao, Xiaohong Sun

**Affiliations:** 1College of Food Science and Technology, Shanghai Ocean University, Shanghai 201306, China; 2Shanghai Engineering Research Center of Aquatic-Product Processing & Preservation, Shanghai 201306, China; 3Laboratory of Quality & Safety Risk Assessment for Aquatic Products on Storage and Preservation (Shanghai), Ministry of Agriculture, Shanghai 201306, China

**Keywords:** bacteriophage, antibiotic-resistant *Salmonella*, biocontrol, genome

## Abstract

*Salmonella enterica* is a major cause of foodborne illness, and the emergence of antibiotic-resistant bacteria has led to huge pressures on public health. Phage is a promising strategy for controlling foodborne pathogens. In this study, a novel *Salmonella* phage vB_SalM_SPJ41 was isolated from poultry farms in Shanghai, China. Phage vB_SalM_SPJ41 was able to lyse multiple serotypes of antibiotic-resistant *S. enterica*, including *S.* Enteritidis, *S.* Typhimurium, *S.* Shubra, *S.* Derby, and *S.* Nchanga. It had a short incubation period and was still active at a temperature <80 °C and in the pH range of 3~11. The phage can effectively inhibit the growth of *S. enterica* in liquid culture and has a significant inhibitory and destructive effect on the biofilm produced by antibiotic-resistant *S. enterica*. Moreover, the phage was able to reduce *S.* Enteritidis and MDR *S.* Derby in lettuce to below the detection limit at 4 °C. Furthermore, the phage could reduce *S*. Enteritidis and *S*. Derby in salmon below the limit of detection at 4 °C, and by 3.9 log_10_ CFU/g and· 2.1 log_10_ CFU/g at 15 °C, respectively. In addition, the genomic analysis revealed that the phages did not carry any virulence factor genes or antibiotic resistance genes. Therefore, it was found that vB_SalM_SPJ41 is a promising candidate phage for biocontrol against antibiotic-resistant *Salmonella* in ready-to-eat foods.

## 1. Introduction

*Salmonella enterica* is one of the greatest threats to human health. Approximately 99% of human salmonellosis is caused by *S. enterica*, with symptoms such as diarrhea, fever, and stomach cramps [1]. It is commonly spread through contaminated food of animal origin, such as dairy products, meat, eggs, and seafood [2,3,4]. Fresh produce can be contaminated with bacterial pathogens at multiple points throughout its production and supply chain [5]. *Salmonella* infection has caused about 115 million illnesses and about 370,000 deaths annually worldwide, and 70–80% of bacterial food poisoning in China is caused by *Salmonella* [6].

However, the overuse of antibiotics on animal-derived products has led to the rise and spread of antibiotic-resistant bacteria (ARB), as well as a series of problems such as drug residues and environmental pollution [7]. Given its tendency to have high rates of adaptation, the acquisition of virulence and resistance genes by *S. enterica* can facilitate the rapid spread of resistant *S. enterica* along the food chain [8]. *S. enterica* isolated from cabbage and lettuce in the Tamale metropolis of Ghana, was resistant to ampicillin (72.22%), erythromycin (94.44%), and ofloxacin (100%) [9]. The World Health Organization has identified antibiotic-resistant *S. enterica* as a critical-priority bacterium [10]. Therefore, new strategies and products for the prevention and control of antibiotic-resistant *S. enterica* need to be studied urgently.

Bacteriophages are viruses that can specifically adsorb and infect bacteria, with a wide range of sources and strong proliferation ability [11]. They have become the focus of attention in recent years due to their efficient bactericidal properties and the advantages of no residue. Phages that specifically eliminate target strains, regardless of target strain resistance, and are not harmful to the normal flora, are an attractive option to replace or complement other existing therapies [12]. The use of bacteriophages to control the growth of *Salmonella* has been reported in various food products, including but not limited to chicken [13], egg fluid [14], milk [15], and lettuce [16]. Furthermore, there have been some commercial phage products on the market, such as the phage product EcoShield^TM^ and Salmonelex, which are allowed to be added to foods to control contaminants such as *Escherichia coli* and *Salmonella* [17].

With the growing demand for food that is both tasty and convenient, ready-to-eat foods are becoming more and more popular. However, ready-to-eat foods may be potential sources of antibiotic-resistant *Salmonella*, posing a threat to food safety. Phages are a promising or effective method of removing pathogenic bacteria from raw food. Therefore, in this study, we isolated a novel phage with lytic activity against antibiotic-resistant *Salmonella* strains, characterized its biology, and analyzed the whole genome. The effect of this phage on the inhibition of *S.* Enteritidis and MDR *S.* Derby in lettuce and raw salmon was further investigated. Our study presented a broad-spectrum phage as a promising antibacterial agent against antibiotic-resistant *Salmonella* in foods. 

## 2. Results

### 2.1. Isolation of Phage SPJ41 and Its Morphology

A lytic phage was isolated from chicken manure using *Salmonella* as a host and named vB_SalM_SPJ41. The morphology of phage SPJ41 observed by transmission electron microscopy is shown in Figure 1A. Phage SPJ41 has an icosahedral head with a diameter of 60.78 nm, a long contractile tail with a length of 110.50 nm, and a width of 17.13 nm. These morphological features indicated that it might be classified as the family *Myoviridae* [18].

### 2.2. Lysis Profiles of Phage SPJ41

The lysis profiles of phage SPJ41 are shown in Table 1. SPJ41 has a broad host spectrum, infecting 11 strains of *S. enterica* (5 serovars) with varying degrees of antibiotic resistance. It failed to lyse any of the non-*Salmonella* strains in this study. SPJ41 was able to completely lyse not only *S.* Typhimurium Sal4 and *S.* Enteritidis ATCC13076, but also multidrug-resistant (MDR) *S.* Derby A174 and *S.* Derby A176. More importantly, the phage could also play an influential lytic role, even though MDR *S.* Derby A63 was resistant to 12 antibiotics and MDR *S.* Derby A64 was resistant to 10 antibiotics.

### 2.3. Optimal MOI and Replication Kinetic

The phage titer was the highest at an MOI of 0.001, which was significantly different from the MOI of 0.01 (*p* < 0.05), indicating that the optimal infection multiplicity of phage SPJ41 was 0.001 (Figure 1B). The growth curve of bacteriophage SPJ41 is shown in Figure 1C. The latent period of SPJ41 was 20 min. The curve can be divided into two phases (from 20 min to 60 min and from 60 min to 120 min). The relative burst size was calculated as approximately 255 PFU/cell in the first stage. 

### 2.4. Stability of SPJ41 at Different pH Values and Temperatures

For the pH stability, there was no noticeable reduction in phage titer at 1 h after exposure to pH 3~11 (Figure 1D). However, when the phage was exposed to an extremely acidic or alkaline environment (pH = 2.0 or pH ≥ 12.0), no phage was detectable. Phage SPJ41 exhibited a high thermal tolerance, as manifested by its stability at 4~50 °C (Figure 1E). When the phage was cultured at 60 °C, the titer decreased slightly by 0.75 log_10_ PFU/mL, and at 70 °C, the phage titer decreased by 1.77 log_10_ PFU/mL. At a high temperature of 80 °C, the phage was undetectable. In general, phage SPJ41 exhibited good temperature tolerance (<80 °C) and wide pH stability (pH 3~11). 

### 2.5. Genomic Characterization of SPJ41

The genome of bacteriophage SPJ41 was sequenced, which revealed a linear dsDNA sequence with 89,584 bp and a GC content of 38.76%. Meanwhile, we found three ncRNAs in the genome of phage SPJ41, including two introns and one sRNA. After the screening, the genome of SPJ41 contained 23 tRNAs, and antibiotic resistance and virulence factor genes were not detected. Meanwhile, transposition and integrase functions that promote horizontal gene transfer were not found in the SPJ4 genome. This genetic background confirms the safety of bacteriophage SPJ41 in food pathogen control and bacteriophage therapy, and that it does not integrate gene fragments into the host bacterial chromosome during its life cycle. The whole genome sequence was uploaded to the GenBank database with the accession number ON868915.1.

A total of 133 open reading frames (ORFs) were identified with 97 ORFs on the positive strand and 38 on the negative strand (Figure 2). The 38 ORFs were assigned functions, which were classified into five functional groups: structural protein module, packaging module, cell lysis module, phage host module, and DNA replication/modification/regulation module. The first structural protein module includes 22 ORFs: tail protein, tail fiber protein, substrate component, tape measure protein, a conserved structural protein, and major capsid protein. The second packaging module contains the terminal enzyme large subunit. The third functional group is the DNA replication recombination and regulation module, which mainly includes 11 ORFs: nucleotide metabolism regulators (deoxynucleotide monophosphate kinases, transcriptional regulatory genes, HNH homing endonucleases, thymidylate synthases, dihydrofolate reductase, anaerobic nucleotide reductase, and glutaredoxin) and DNA replication(DNA primerase/helicase, DNA polymerase, and DNA ligase). Endolysin and o-spanin belong to the fourth functional group, which is the cell lysis module. The last functional group for phage–host interactions includes the rIIa and rIIB proteins [19].

### 2.6. Classification and Genome Comparison of Lytic Phage SPJ41

When compared to all sequenced phages on NCBI, SPJ41 shared the highest nucleotide identity (94.00% coverage and 95.99% identity) with *Salmonella* phage BPSELC-1 (accession no. MN227145). BPSELC-1 is a member of the genus *Felixounavirus* of the subfamily *Ounavirinae* [20]. According to the International Committee on Taxonomy of Viruses (ICTV) classification [21], phage sequences of different genera of the *Ounavirinae* subfamily were downloaded from NCBI, and a phylogenetic tree was constructed with terminal enzyme large subunit and major capsid protein. Both phylogenetic trees showed that SPJ41 was classified into the *Ounavirinae* subfamily, *Felixounavirus* genus (Figure 3). To further compare the differences between phage SPJ41 and members of the genus *Felixounavirus*, a comparative genome circle diagram based on the Brig software is shown in Figure 4. The phage genomes of the genus *Felixounavirus* showed high similarity and only showed differences in some hypothetical protein regions and four functional protein regions. In the 50k bp region, these phage tail fiber proteins exhibited divergence, which correlates with the phage host range. In addition, there are differences in the three homing endonuclease (HNH endonuclease) gene sequence regions, which are site-specific enzymes for breaking DNA double strands and play a regulatory role in transcription.

### 2.7. Inhibition Curves of Phage SPJ41 on Salmonella

The inhibitory effect of phage SPJ41 on *S.* Typhimurium Sal4, *S.* Enteritidis ATCC13076, and MDR *S.* Derby A63 was evaluated in LB broth. The result is shown in Figure 5. The bacterial cell densities (OD_600_) of *S.* Typhimurium Sal4 and *S.* Enteritidis ATCC 13076 remained less than 0.2 until 6 h, and until 8 h for MDR *S.* Derby A63. In the phage treatment group, except for the MOI of 0.01, the measured OD_600_ value was always lower than 0.1 in the first 5 h. After incubation for 24 h, the phage-treated group with an MOI of 10,000 had the lowest bacterial cell density, which was significantly different from the other treated groups (*p* < 0.01).

### 2.8. Inhibition and Disruption of Biofilms by SPJ41 Phage Treatment

Phage SPJ41 exhibited significant inhibitory and disruptive effects on the biofilm of *S.* Enteritidis ATCC13076 and MDR *S.* Derby A63 at different MOIs (Figure 6). Phage SPJ41 inhibited *S.* Enteritidis ATCC13076 from forming biofilms by approximately 6% to 53%. For MDR S. Derby A63, it was able to inhibit biofilm formation by 15% to 43% (Figure 6A). Biofilms formed by *S.* Enteritidis ATCC13076 were treated with phage SPJ41, and a 33% to 60% disruption in the attached biofilm was recorded. For MDR *S.* Derby A63, the attached biofilm was disrupted by 37% to 74% (Figure 6B).

### 2.9. Phage SPJ41 Biological Control of Salmonella in Two Ready-to-Eat Foods

The effect of phage SPJ41 on reducing *S.* Enteritidis ATCC13076 and MDR *S.* Derby A63 on lettuce is shown in Figure 7A,B. Phage treatment reduced *S.* Enteritidis ATCC13076 to below the detection limit (10 CFU/cm^2^) at 3 h at 4 °C. However, for MDR *S.* Derby A63, there was a reduction of 1.9 log_10_ CFU/cm^2^ at 3 h and below the detection limit at 9 h (Figure 7A). Compared to the control, *S.* Enteritidis ATCC13076 and MDR *S.* Derby A63 on lettuce with phage SPJ41 treatment reduced 3.2 log_10_ CFU/cm^2^ and 2.5 log_10_ CFU/cm^2^ at 25 °C for 24 h (Figure 7B).

The results of phage SPJ41 reducing *S.* Enteritidis ATCC13076 and MDR *S.* Derby A63 on salmon are shown in Figure 7C,D. A reduction of approximately 2.2 log_10_ CFU/g of *S.* Enteritidis ATCC13076 was measured in the treated group compared to the control, incubated at 4 °C for 24 h. For MDR *S.* Derby A63, a decrease of approximately 0.7 log_10_ CFU/g and 1.1 log_10_ CFU/g was measured for 3 h and 24 h incubation at 4 °C, respectively (Figure 7C). *S.* Enteritidis ATCC 13076 on salmon treated with phage SPJ41 had viable counts below the detection limit (50 CFU/g) at 15 °C for 6~12 h. After treatment for 24 h, *S.* Enteritidis ATCC13076 decreased by 3.9 log_10_ CFU/g, exhibiting a remarkably significant bacteriostatic effect (*p* < 0.0001) at 15 °C. MDR *S.* Derby A63 in phage-treated salmon after 3 h and 24 h decreased by 1.5 log_10_ CFU/g and 2.1 log_10_ CFU/g, respectively, which was significantly lower than the control at 15 °C (Figure 7D).

## 3. Discussion

In the fight against antibiotic-resistant pathogens, there is renewed interest in phages as alternative or complementary antimicrobials [22,23,24]. In this study, a lytic phage SPJ41, subjected to a member of the genus *Felixounavirus* of the *Ounavirinae* subfamily, was isolated from chicken manure. The result of TEM showed that the morphological features of SPJ41 were similar to those of other members of the genus *Felixounavirus*, such as Felix O1 (73 nm diameter icosahedral head and 17 × 113 nm diameter retractable tail) and BPSELC-1 (about 83.3~91.6 nm diameter nm head and 16 × 116.6 nm diameter tail) [20,25]. The phages were observed to aggregate together, which was presumably attributable to the low sodium concentration in the phage medium [26]. Previous reports have shown that phages of the genus *Felixounavirus*, such as SP116 and Felix O1, were capable of infecting several different serovars of *Salmonella* [27,28]. The phage SPJ41 isolated in this study was also able to lyse five *Salmonella* serotypes, including *S.* Typhimurium, *S.* Enteritidis, *S.* Shubra, *S.* Derby, and *S.* Nchanga. And most of these strains were antibiotic-resistant *Salmonella*. In particular, the phage SPJ41 can lyse *S.* Derby A63, resistant to 12 kinds of antibiotics, which was increasing in fresh foods [29]. This indicated that the phage SPJ41 was a promising candidate to control the MDR *S. enterica.*

Phages with short incubation periods are more suitable for biocontrol because they can lyse more bacteria in a certain time [30]. The replication kinetic curve of phage SPJ41 was determined with an optimal MOI of 0.001, and the results showed that the latent period was 20 min, and the burst size was 255 PFU/cell. The latent period was similar to that of phage Felix O1 (20 min), but the burst size was larger than that of Felix O1 [28]. Phage SPJ41 exhibited good temperature tolerance (<80 °C) and broad pH stability (pH 3~11). Compared with the *Salmonella* phage reported previously [14,31], phage SPJ41 showed higher tolerance to an extreme environment, being naturally resistant to harsh physicochemical environmental influences, making it more beneficial as a biocontrol agent in food and processing environments [32].

Based on morphological observation and genome comparison, SPJ41 belongs to the subfamily *Ounavirinae*. Phage SPJ41 is a linear dsDNA sequence with a genome size of 89,584 bp. The genome sizes found here were in the range reported by the *Felixounavirus* genus, which described a wide range of sizes from approximately 83,000 to 90,000 bp [33]. In the genome of SPJ41, endolysin, o-spanin, and dihydrofolate were found. Dihydrofolate reductase reduces 7,8-dihydrofolate to tetrahydrofolate and acts as a cofactor in the conversion of dUMP to dTMP by thymidylate synthase enzyme [34]. Endolysin and o-spanin belong to the fourth functional group, the cell lysis module. Spanins are phage lysis proteins that act together to form a bridge between i-spanin and o-spanin [35]. While only o-spanin genes were found in SPJ4, further studies are needed to understand the function of spanins in these phages. The SPJ41 genome contains 23 tRNAs, and studies have shown that tRNAs play an important role in the synthesis of phage coat and tail proteins [36]. The genetic background confirms the safety of bacteriophage SPJ41 in food pathogen control and bacteriophage therapy, and it does not integrate gene fragments into the host bacterial chromosome during its life cycle.

When compared to all sequenced phages on NCBI, SPJ41 shared the highest nucleotide identity (90.23%) with *Salmonella* phage BPSELC-1. According to the ICTV [21], the main species classification standard for bacterial and archaeal viruses is 95% genome similarity. Thus, SPJ41 isolated in this study was a new species of *Felixounavirus*. The genomes revealed a high similarity in the phage genomes of the genus *Felixounavirus* according to the mapping of the comparative genomic circles. However, phages of the genus *Felixounavirus* showed considerable diversity in the tail fiber protein because the bacterial receptor of this genus corresponds to liposaccharides, which is a molecule with high variability [33,37]. Barron-Montenegro [33] found that the genomes of phages of the genus *Felixounavirus* distributed in different geographical locations are highly conserved, while the tail fibers show considerable diversity. Interestingly, we also found differences in the HNH homing endonuclease, which plays a regulatory role in transcription. HNH homing endonucleases are site-specific enzymes that disrupt DNA duplexes, allowing insertion or mobilization of genes, and play an important role in the evolution of the *Siphovirus* genome [25,33]. 

Ready-to-eat food is a potential source of antibiotic-resistant *Salmonella* that poses a threat to food safety. In addition to *S*. Enteritidis, *S*. Derby serovars have recently been found to be more common [29]. Phages have previously been used to control antibiotic-resistant *Salmonella* on lettuce [15], milk [38], and egg fluid [14]. In this study, phage SPJ41 was able to reduce *S.* Enteritidis ATCC13076 and MDR *S.* Derby A63 in lettuce to below the detection limit at 4 °C by 3.2 log_10_ CFU/cm^2^ and 2.5 log_10_ CFU/cm^2^ at 25 °C, respectively, which was nearly three times higher than that reported by Guo et al. [39]. Phage SPJ41 reduced *S.* Enteritidis ATCC13076 and MDR *S*. Derby A63 on salmon by about 2.2 log_10_ CFU/g and 1.1 log_10_ CFU/g at 4 °C, respectively. Finally, at 15 °C, it reduced *S.* Enteritidis ATCC13076 and MDR *S*. Derby A63 on salmon by 3.9 log_10_ CFU/g and 2.1 log_10_ CFU/g, respectively. At the inoculum level of 10^4^ CFU/g in salmon, phage SLMP1 at a dose of 10^8^ PFU/g could reduce approximately 1.5 to 2.5 log CFU/g of *Salmonella* counts compared with the control [40]. Therefore, phage SPJ41 was effective in controlling antibiotic-resistant *S. enterica* in ready-to-eat foods. 

## 4. Materials and Methods 

### 4.1. Bacteria Strains and Culture Conditions

A total of 12 antibiotic-resistant *S. enterica* strains representing five serotypes (*S.* Enteritidis, *S.* Typhimurium, *S.* Shubra, *S.* Derby, and *S.* Nchanga) and six non-*Salmonella* bacterial strains were used in this study (Table 1). A total of 12 antibiotic-resistant *S. enterica* exhibited varying degrees of antibiotic resistance, nine of which were antibiotic-resistant strains. *S.* Derby A63 was the most resistant, and it was resistant to 12 antibiotics (AMP, CFZ, TET, DOX, CHL, CIP, OFX, SMZ, ERY, AZM, GEN, and KAN). *S.* Derby A64 was resistant to 11 antibiotics (AMP, TET, DOX, CIP, OFX, SMZ, ERY, AZM, KAN, and GEN). All strains were used for phage lytic range determination. *S.* Typhimurium Sal4 was used as a host strain for phage isolation, propagation, and purification. To carry out bacterial cultures, each strain was cultured by picking an isolated colony from Luria–Bertani agar (LB, Land Bridge Technology, Beijing, China) plate, and then inoculated into Luria–Bertani broth at 37 °C overnight with agitation at 200 rpm. 

### 4.2. Bacteriophage Isolation and Purification

Phage was isolated from chicken stool in Shanghai, China, using the method described previously by Cao et al. [41]. Briefly, 25 g of sample was mixed homogeneously with 40 mL of LB broth, supplemented with 350 μL of 1 M CaCl_2_, and inoculated with 1 mL of *Salmonella* suspension (10^9^ CFU/mL). After incubation at 37 °C overnight, the medium was centrifuged at 8000 rpm for 15 min (5424, Eppendorf AG 22331, Hamburg, Germany). The supernatant was filtered with a 0.22 μm pore (Millipore, Billerica, MA, USA) size syringe filter, and the presence of lytic phage in the sample was confirmed by spot testing. A district plaque was observed and suspended in 1 mL of salt magnesium (SM) buffer for purification. The phages were purified at least three times to create phage stock, using a double-layer agar technique [42]. 

### 4.3. Determination of Phage Host Range

The host range of the isolated phage was determined by the spot test method as described elsewhere with some modifications [27]. Ten microliters of a suspension containing phage particles (10^9^ PFU/mL) were dropped on the surface of lawn cultures of selected 12 *S. enterica* and six non-*Salmonella* strains. The plates were observed for the appearance of the clear zone after incubation at 37 °C overnight.

### 4.4. Transmission Electron Microscopy (TEM) of SPJ41

The enriched phages were purified by cesium chloride density gradient centrifugation. Twenty microliters of purified phage solution (10^9^ PFU/mL) was added dropwise to copper mesh and fixed for 10 min, and the residual liquid was absorbed by filter paper. Then, 2% phosphotungstic acid was added to the stain for 2 min. The sample was dried in a sterilized biosafety cabinet and observed by a transmission electron microscope (Philips, Eindhoven, The Netherlands).

### 4.5. Optimal Multiplicity of Infection (MOI) Determination

The MOI of the phage was determined using the method described previously by Li et al. with minor modifications [20]. Briefly, the phage and its host cells were mixed at 10:1, 1:1, 1:10, 1:100, 1:1000, 1:10,000, and 1:100,000, respectively. Then, the mixture was added to 5 mL of fresh LB medium and incubated with shaking at 37 °C for 4 h. The culture was centrifuged at 10,000 rpm (5424, Eppendorf AG 22331, Hamburg, Germany) for 5 min and filtered using 0.22 µm filters. The phage titer was determined using the double-layer agar method. The group with the highest titer was the optimal MOI for phage.

### 4.6. Replication Kinetic Curve of Phage SPJ41

A one-step growth curve experiment was performed according to the method described by Yi et al. with modifications [43]. Briefly, *S.* Typhimurium Sal4 was infected with the phage at the optimal MOI of 0.001 and incubated at 37 °C for 15 min without shaking. The mixture was centrifuged at 4000 rpm for 15 min in a bench-top centrifuge (5810R, Eppendorf, Avanti J-26XP, Germany) and the supernatant was discarded. The bacteria–phage pellet was washed with SM buffer and resuspended in 10 mL of fresh LB, followed by incubation at 37 °C with shaking at 200 rpm. Samples were taken at 10 min intervals, then immediately diluted, and plated for phage titer quantification. Relative burst size = (phage titer at the end of the burst cycle − initial phage titer)/initial phage titer.

### 4.7. pH and Temperature Stability of Phage SPJ41

To evaluate the pH stability of phage SPJ41, 100 μL of phage suspension (10^9^ PFU/mL) was transferred into 900 μL of SM buffer at different pH values (pH 2–13, adjusted using NaOH or HCl). The phage suspensions were incubated at 37 °C for 1 h. To access the temperature stability of phage SPJ41, 100 μL of the phage suspensions (10^8^ PFU/mL) were incubated at 4, 15, 25, 37, 50, 60, 70, and 80 °C for 60 min. To measure the pH or temperature stability of phage SPJ41, the phage titers were tested after incubating using the double-layer agar plate method.

### 4.8. Phage SPJ41 Genome Sequencing, Annotation, and Comparison

Phage DNA was subjected to phenol/chloroform DNA extraction [41]. Libraries of different inserts were constructed using the whole-genome shotgun (WGS). Paired-end (PE) sequencing of these libraries was performed on the Illumina NovaSeq sequencing platform using next-generation sequencing (NGS). De novo assembly of high-quality next-generation sequencing data was performed using A5-miseq v20160825 [44] and SPAdesv3.12.0 [45] to construct contig sequences. GeneMarkS v4.32 (http://topaz.gatech.edu/GeneMark/) [46] software was used to perform gene prediction on the whole gene sequence, and protein-coding genes were functionally annotated based on the non-redundant database (NR) on NCBI. The tRNAscan-SE [47] was used to predict tRNA genes in the genome. The Antibiotic Resistance Database (ARDB, https://card.mcmaster.ca/analyze/rgi accessed on 1 April 2022) and Virulence Factor Database (VFDB http://www.mgc.ac.cn/VFs/ accessed on 1 April 2022) were used to analyze antibiotic resistance and virulence factor, respectively. Phage gene maps were constructed using GCview server [48]. 

Phylogenetic analyses of the phage major capsid protein and terminase large subunit were also performed using MEGA 7 [14] and further refined using the website evolview (https://evolgenius.info//evolview-v2/#login accessed on 20 June 2022). The overall DNA sequence homolog was defined as coverage multiplied by identity according to the International Committee on Taxonomy of Viruses (ICTV) [21]. Subsequently, phage SPJ41 and other phages of the genus *Felixounavirus* were compared and visualized with highly similar specific sequence fragments using BRIG, with parameters set by default. 

### 4.9. Inhibition Curve of Bacteriophage SPJ41 against Salmonella 

To examine the inhibitory activity of phage SPJ41, we incubated the phage with liquid cultures of *S.* Typhimurium Sal4, *S.* Enteritidis ATCC 13076, and MDR *S.* Derby A63 at MOIs of 0.01, 1, 100, and 10,000 for 24 h. An amount of 100 μL of suspension (10^6^ CFU/mL) of bacterial broth was mixed with 100 μL of phage SPJ41 lysate at different MOIs. The experimental group without phage and only bacterial culture was set as the control group, and the phage and bacterial culture were not added to the blank group. At 0, 2, 4, 6, 8, 10, 12, and 24 h, the OD_600_ value was measured by a microplate reader to monitor bacterial growth.

### 4.10. Effect of Phage SPJ41 against Salmonella Biofilms 

Phage SPJ41 inhibited and disrupted *S*. Enteritidis ATCC 13076 and MDR *S*. Derby A63 biofilms as previously described in García et al. [49] and Xie et al. [50] with minor modifications. To inhibit the formation of biofilm, 100 μL of *Salmonella* suspension (10^6^ CFU/mL) was added to a 96-well polystyrene microplate, followed by 100 μL of phage liquid at different MOIs (10,000, 100, 1, and 0.01) and incubated at 37 °C for 24 h. For the disrupted biofilm, 200 μL (1 × 10^6^ CFU/mL) of *Salmonella* was added to sterile 96-well plates and incubated at 37 °C for 24 h to form biofilm. The planktonic cells were washed with PBS, and a liquid medium containing different concentrations of phage (1 × 10^7^, 1 × 10^8^, 1 × 10^9^, and 1 × 10^10^ PFU/mL) was added and incubated at 37 °C for 24 h. The cultures were removed, and the 96-well plates were washed three times with PBS and placed in an oven to air dry at 40 °C for 30 min. Then, 200 μL of 0.1% crystal violet was added and incubated at 37 °C for 30 min. Excess crystal violet was washed off. An amount of 200 μL of absolute ethanol was added and dissolved at 37 °C for 5 min, and the OD_570_ value was measured with a microplate reader. 

### 4.11. Effect of Phage SPJ41 against Salmonella in Food Samples

Lettuce and salmon, purchased from the Nong Gong Shang Supermarket in Shanghai, China, were sliced aseptically in the laboratory. Lettuce and salmon samples were inoculated with *S.* Enteritidis ATCC13076 and MDR *S.* Derby A63 followed by phage addition described by Xu et al. [19] and Guo et al. [39,40]. 

The inner leaves of the lettuce were disinfected with 75% ethanol and UV-treated, then cut into a 1 cm × 1 cm square using a sterile knife. Sterility was ensured by cultivation on TSA agar. The lettuce samples were inoculated with 10 μL of *Salmonella* suspension to achieve a viable count of approximately 1 × 10^4^ CFU/cm^2^ and left at room temperature for 10 min. Then, 10 μL of phage solution was added to the final titer to 1 × 10^8^ PFU/cm^2^. PBS buffer instead of phage fluid was used as a control. Aliquots were extracted after 0, 3, 6, 9, 12, and 24 h of incubation at 4 °C and 25 °C and suspended in 1 mL sterile PBS buffer solution. The suspension samples were homogenized for 2 min, and then the viable bacterial count (CFU/mL) was determined by serial dilution plate counting.

Fresh salmon meat was aseptically cut into slices (2 × 2 cm, ~1 g). Subsequently, 20 μL of *Salmonella* suspension (1 × 10^6^ CFU/mL) was inoculated onto the surface of the salmon fillets and adhered at room temperature for 10 min. An amount of 20 μL of phage SPJ41 (1 × 10^10^ CFU/mL) was added to the salmon fillets to make a final dose of 2 × 10^8^ PFU/g. All samples were prepared in triplicate after incubation at 4 °C and 15 °C for 0, 3, 6, 9, 12, and 24 h, collected and resuspended in 5 mL PBS. The suspension was vortexed and homogenized for 2 min, and the number of viable bacteria (CFU/mL) was determined.

### 4.12. Statistical Analysis

All experiments in this study were repeated three times. The data are expressed as mean ± standard deviation (SD) and the differences were analyzed with two-way ANOVA using GraphPad Prism 9.0. Differences were considered statistically significant at *p* < 0.05. 

## 5. Conclusions

In this study, we isolated and characterized a novel lytic phage vB_SalM_SPJ41 from chicken manure. Phage SPJ41 tolerates high temperatures and extreme pH, and exhibits a broad lysis profile (11/12). Genome comparison revealed that phage SPJ41 is a new member of the genus *Felixounavirus* of the subfamily *Ounavirinae*. In addition, the phage does not carry any virulence factor genes or antibiotic resistance genes. It can effectively control the growth of antibiotic-resistant *S. enterica* in two ready-to-eat foods. Therefore, phage SPJ41 can be used as a candidate biocontrol agent to inhibit antibiotic-resistant *S. enterica* in the processing and preservation of ready-to-eat foods.

## Figures and Tables

**Figure 1 antibiotics-12-00364-f001:**
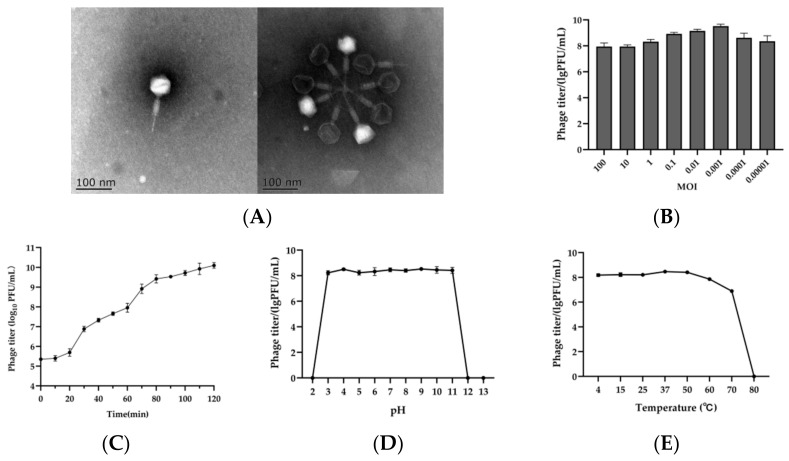
Biological characteristics of phage SPJ41. (**A**) Transmission electron micrographs of SPJ41. (**B**,**C**) Optimal MOI and replication curve of phage SPJ41. (**D**,**E**) Stability of SPJ41 at different pH values and temperatures.

**Figure 2 antibiotics-12-00364-f002:**
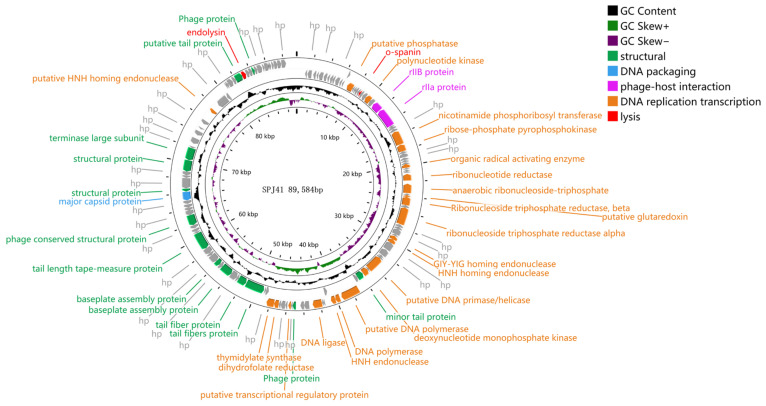
Circular genome annotation of SPJ41. Circular genome maps of SPJ41 using GCview server. The rings from the inside out represent GC skew (green and purple), GC content (black), and CDS (blue). The different colored arrows represent the different functions of the predictive open reading frame (ORF): red, cell lysis module; gray, hypothetical protein; orange, DNA replication/modification/regulation; green, phage structure; blue, DNA packaging; and purple, phage–host interactions.

**Figure 3 antibiotics-12-00364-f003:**
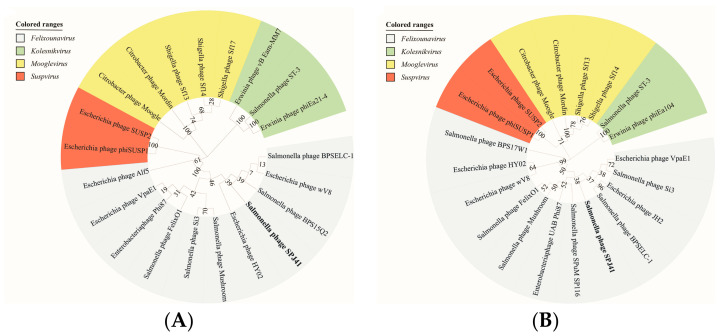
Phylogenetic analyses of SPJ41. (**A**,**B**) Phylogenetic analyses of selected phages and phages of the proposed new genus based on the protein sequence of terminase large subunit and major capsid protein. Members of the genera *Felixounavirus*, *Kolesnikvirus*, *Mooglevirus,* and *Suspvirus* virus were illustrated with sky blue, pale green, yellow, and red, respectively.

**Figure 4 antibiotics-12-00364-f004:**
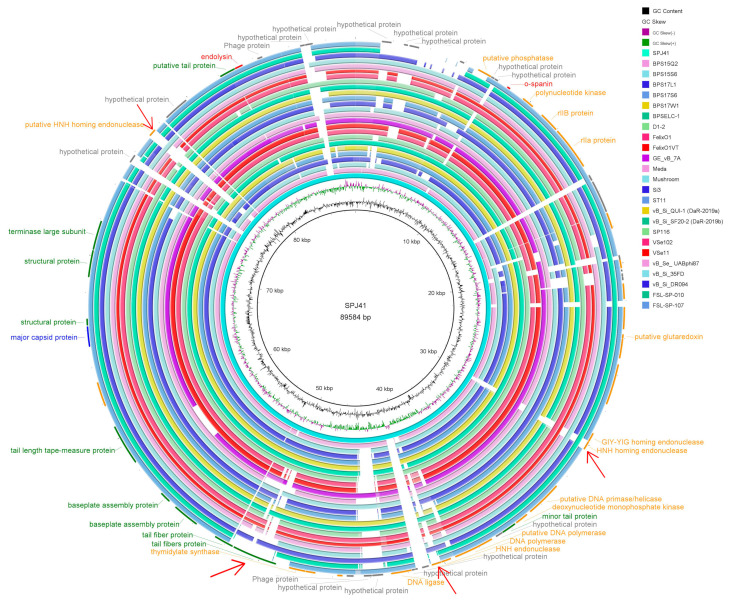
The phage comparative genome circle map of 25 strains of the genus *Felixounavirus*. From the inner circle to the outer circle are GC content, GC skew, reference genome SPJ41, and the phages of the genus *Felixounavirus*. The outermost circle is the functional annotation: red, cell lysis module; gray, hypothetical protein; orange, DNA replication/modification/regulation; green, phage structure; blue, DNA packaging; and purple, phage–host interactions. The red arrows are functional protein difference regions.

**Figure 5 antibiotics-12-00364-f005:**
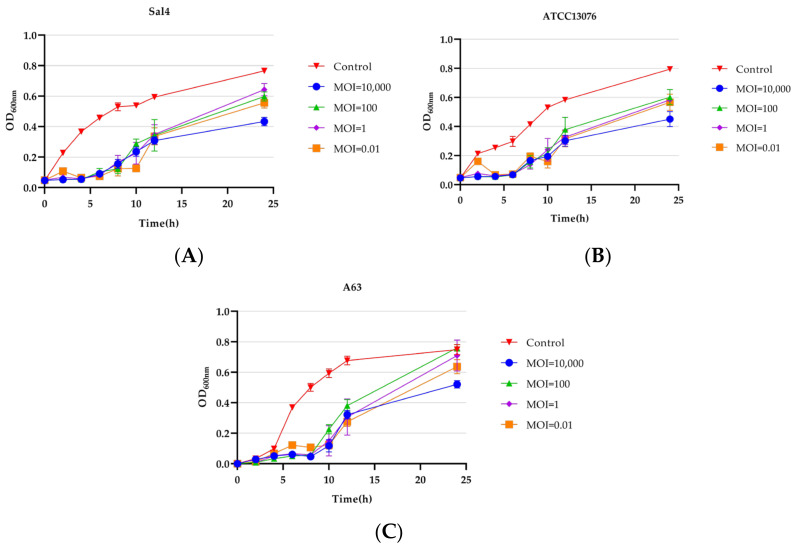
Phage SPJ41 inhibition of *S.* Typhimurium Sal4 (**A**), *S.* Enteritidis ATCC 13076 (**B**), and MDR *S.* Derby A63 (**C**).

**Figure 6 antibiotics-12-00364-f006:**
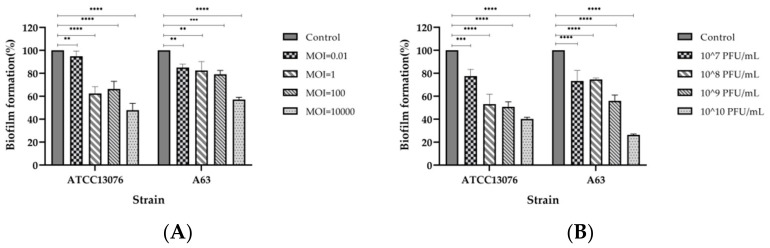
(**A**) Effect of phage SPJ41 on inhibition of biofilm formation of *S.* Enteritidis ATCC 13076 and MDR *S*. Derby A63. (**B**) Effect of phage SPJ41 on the removal of biofilms of *S.* Enteritidis ATCC 13076 and MDR *S*. Derby A63. Statistical comparisons were performed relative to the control using the two-way ANOVA with multiple comparisons test (**, *p* < 0.01; ***, *p* < 0.001; ****, *p* < 0.0001).

**Figure 7 antibiotics-12-00364-f007:**
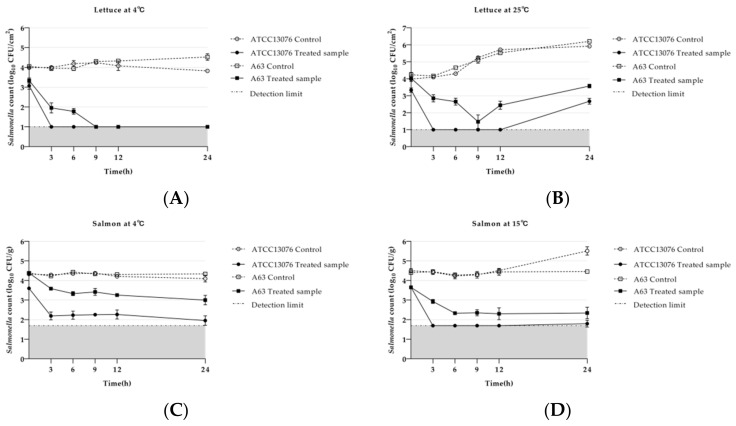
Effect of phage SPJ41 on reducing *Salmonella* in lettuce. (**A**,**B**) Effect of phage SPJ41 on reducing *S*. Enteritidis ATCC13076 and MDR *S*. Derby A63 on lettuce at 4 °C and 25 °C. (**C**,**D**) Effect of phage SPJ41 on reducing *S*. Enteritidis ATCC13076 and MDR *S*. Derby A63 in salmon at 4 °C and 15 °C.

**Table 1 antibiotics-12-00364-t001:** Phage SPJ41 lysis profiles.

Strains	Source	Drug Resistance ^d^	Lytic ^e^
*Salmonella* Enteritidis			
ATCC13076	ATCC ^a^	ERY	+++
A184	Customs ^b^	AMP, CFZ, FEP, TET, DOX, CIP, OFX, SMZ, ERY, AZM, KAN	++
*Salmonella* Typhimurium			
Sal4	Customs	AMP, ERY	+++
A18	Customs	AMP, DOX, SMZ, ERY, GEN	
*Salmonella* Shubra			
Sal6	Customs	AMP, TET, CHL, SMZ, ERY	++
*Salmonella* Derby			
Sal18	Customs	AMP, DOX, SMZ, ERY	+
A63	Customs	AMP, CFZ, TET, DOX, CHL, CIP, OFX, SMZ, ERY, AZM, GEN, KAN	++
A64	Customs	AMP, TET, DOX, CIP, OFX, SMZ, ERY, AZM, KAN, GEN	++
A174	Customs	AMP, TET, DOX, SMZ, ERY	+++
A176	Customs	AMP, TET, DOX, SMZ, ERY	+++
*Salmonella* Nchanga			
A91	Customs	AMP, TET, DOX, CHL, SMZ, ERY, AZM, GEN	+
A92	Customs	AMP, TET, DOX, CHL, SMZ, ERY, AZM, GEN	+
*Listeria monocytogenes*			
ATCC19115	ATCC		−
ATCC19116	ATCC		−
CMCC25926	CMCC ^c^		−
*Escherichia coli*			
ATCC25922	ATCC		−
*Staphylococcus aureus*			
ATCC29213	ATCC		−
CMCC26003	CMCC		−

^a^ ATCC, American Type Culture Collection. ^b^ Customs, Preserved by Shanghai Customs Animal, Plant, and Food Inspection and Quarantine Technology Center. ^c^ CMCC, Shanghai Preserved Biology Company. ^d^ AMP, ampicillin; CFZ, cefazolin; FEP, cefepime; TET, tetracycline; DOX, doxycycline; CHL, chloramphenicol; CIP, ciprofloxacin; OFX, doxycycline; SMZ, sulfamethoxazole; ERY, erythromycin, AZM, azithromycin; GEN, gentamicin; KAN, kanamycin. ^e^ Lytic, +++, complete lysis; ++, lysis; +, turbid lysis; −, no plagues.

## Data Availability

Sequences generated in this study are publicly available at ncbi. The data presented in this study are available on request from the corresponding author. The data are not publicly available.

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
