# Peer review of "Characterization of Phage vB_SalM_SPJ41 and the Reduction of Risk of Antibiotic-Resistant Salmonella enterica Contamination in Two Ready-to-Eat Foods"

_antibiotics, 2023, doi:10.3390/antibiotics12020364_

Round 1

Reviewer 1 Report

General comments

The paper is well written, is innovative and provides valuable information dealing with the characterization of a novel antibacterial lytic phage vB_SalM_SPJ41 from chicken manure, with a promising potential use to control the growth of antibiotic-resistant Salmonella in ready-to-eat foods. In addition, the main objectives of the work are clearly declared, the results are appropriately discussed and the conclusions are supported by the results obtained in the work.

However, there are some points that should be addressed by the authors to improve the presentation and understanding of the paper.

Other considerations are as follows:

Abstract

1.1. Line 66: S. Enteritidis and S. Derby should be written in italics, with the species in lowercase letters. Please correct also throughout the manuscript.

1. Introduction section

1.1. Lines 18–19, 22, 23: S. Enteritidis, S. Typhimurium, S. Shubra, S. Derby and S. Nchanga should be written in italics, with the species in lowercase letters. Please correct also throughout the manuscript.

2. Results section

2.1. Lines 103–106 and 256–257: As observed in Figure 1C, the phage titer increased from 20 min describing two cycles of growth (from 20 min to 60 min and from 60 min to 120 min) like a diauxic growth pattern. Please provide a clear explanation for this observation. Did other researchers observe before this behavior? Please explain and provide references to support this observation.

2.2. Lines 108–115 and 258–262: The phage titer showed good stability after exposure for 1 h at pH 3~11 (Figure 1D) and at 4°C~50°C (Figure 1E). Why did the phage titer experiment the abrupt decrease at pH values 2 and 12 or at 80 ºC, being not detected in these conditions?

2.3. Figure 5: The Figure caption should be shortened and corrected as follows: Figure 5. Phage SPJ41 inhibition of S. typhimurium Sal4 (A), S. enteritidis ATCC 13076 (B) and MDR S. derby A63 (C).

2.4. Figure 5: In Figure 5B, the growth of S. enteritidis ATCC 13076 (MOI = 0.01) showed an atypical increase at 2 h of incubation and then decreased at 4 h. Please explain why? The same for S. typhimurium Sal4 (Figure 5A) and MDR S. derby A63 (Figure 5C).

2.5. Line 182: “for 6 h, 6 h, 8 h, …..”. I don’t understand, please correct.

2.6. Line 183: Replace “the phage treated group with an MOI of 10000” with “the phage-treated group with a MOI of 10000”.

2.7. Figure 6: The Figure caption should be corrected. I didn’t find the notation ns and asterisk (*). Additionally, the meaning of P , 0.01; ***, P , 0.001; ****, P , 0.0001 is unclear.

2.8. Line 213: Replace “24 hours” with “24 h”.

2.9. Figure 7: In Figure 7B, in the treated lettuce samples at 25ºC, the growth of S. enteritidis ATCC 13076 and MDR S. derby A63 showed an increase from 9 and 12 h, respectively until the end of the incubation (24 h). This behavior was not observed in the salmon samples at 25ºC. Please explain why?

2.10. Figure 7: The bacterial counts in the y-axis in Figures 7A, B, C and D are represented as log10 CFU/Sample. I think that the y-axis should be corrected as log10 CFU/g.

3. Discussion section

3.1. Lines 233–238: This information should be moved to the Introduction section since it is not represent a discussion of results.

4. Materials and Methods section

4.1. Lines 363–370: Why the pH stability of phage SPJ41 was not conducted using buffers instead of NaOH or HCl?

4.2. Lines 363–370: Why different volumes (100 µL or 200 µL) and phage suspensions (109 PFU/mL or 108 PFU/mL) were used to evaluate the pH and temperature stability?

4.3. Lines 363–370: Why didn´t the authors perform the evaluation of the pH and temperature stability using the response surface methodology? With this approach, the true optimum values for pH and T could be obtained and the number of experiments could be considerably reduced.

4.4. Lines 391–399: Why didn´t the authors calculate the lethal dose (LD50) of phage SPJ41 against S. typhimurium Sal4, S. enteritidis ATCC 13076 and MDR S. derby A63? With the data measured in this assay (OD600 values), the corresponding inhibition curves and LD50 of phage SPJ41 for each strain could be obtained.

4.5. Line 403: Replace “CFU/ml” by “CFU/mL”.

4.6. Line 425: Replace “24 hours” by “24 h”.

4.7. Line 438: Please provided the year of the GraphPad Prism 9.0 program used.

4.8. Lines 438–439: Replace “A p value of < 0.05 was considered as statistically significant” by “Differences were considered statistically significant for p < 0.05”.

Author Response

Response to Reviewer 1 Comments

Dear Reviewer:

Thank you for your reply concerning our manuscript entitled “Characterization of phage vB_SalM_SPJ41 and reduction of risk of antibiotic-resistant Salmonella enterica contamination in two ready-to-eat foods” (ID: antibiotics 2174202). Those comments are all valuable and helpful for improving our paper. We have revised the manuscript according to the comments and highlighted the revised portion (red font is the modified content). The main correction and the respond to reviewers’ comments are as follows:

Point 1: Abstract: 1.1. Line 66: S. Enteritidis and S. Derby should be written in italics, with the species in lowercase letters. Please correct also throughout the manuscript.

Response 1: Thank you for your comments. “Enteritidis” and “Derby” in the manuscript are serotypes of Salmonella enterica, not species. So "Enteritidis" and "Derby" are not written in italics and start with a capital letter.

Point 2: 1. Introduction section: 1.1. Lines 18–19, 22, 23: S. Enteritidis, S. Typhimurium, S. Shubra, S. Derby and S. Nchanga should be written in italics, with the species in lowercase letters. Please correct also throughout the manuscript.

Response 2: Thank you for your comments. “Enteritidis”, “Typhimurium”, “Shubra”, “Derby” and “Nchanga” in the manuscript are serotypes of Salmonella enterica, not species. So “Enteritidis”, “Typhimurium”, “Shubra”, “Derby” and “Nchanga” are not written in lowercase letters and italics.

Point 3: 2. Results section: 2.1. Lines 103–106 and 256–257: As observed in Figure 1C, the phage titer increased from 20 min describing two cycles of growth (from 20 min to 60 min and from 60 min to 120 min) like a diauxic growth pattern. Please provide a clear explanation for this observation. Did other researchers observe before this behavior? Please explain and provide references to support this observation.

Response 3: Thank you for your comments. Most phages follow a one-step growth curve pattern. The Salmonella phages (OSY-STA and OSY- SHC) reported by Yi et al. is similar to the phage vB_SalM_SPJ41 in replication pattern [1]. Yi et al. called it “replication kinetic” instead of “one-step growth curve”. The burst size of phage OSY-STA and OSY-SHC was only calculated for the first stage. So, we have corrected the “one-step growth curve” to “replication kinetic” in the Results section: 2.3, Materials and methods: 4.6 and Discussion section. We emphasized in the Results section: 2.3 that the growth curves are in two phases, and the first phase burst size was calculated. We have also modified the calculation of burst size in the Methods section: 4.6 as “Relative burst size = (final phage titer – initial phage titer) / initial phage titer”.

Point 4: 2.2. Lines 108–115 and 258–262: The phage titer showed good stability after exposure for 1h at pH 3~11 (Figure 1D) and at 4°C~50°C (Figure 1E). Why did the phage titer experiment the abrupt decrease at pH values 2 and 12 or at 80 ºC, being not detected in these conditions?

Response 4: Thank you for your comments. In our study, phage SPJ41 was treated at at 4°C, 15°C, 25°C, 37°C, 50°C, 60°C, 70°C, 80°C for 60 min and pH 2-13. Park et al. [2] treated Salmonella phage MSP1 at different temperatures (25, 50, 55, 60, 65 and 70°C) and measured the phage titer every 10 min for 60 min. The results showed that phage MSP1 titer decreased with the increase of treatment time and temperature. The reasons for the abrupt decrease of phage titer after treatment at 80°C for 1 h are probably as follows: (1) The phage titer was only measured after treatment for 1 h. It was not measured within 1 hour. (2) The temperature intervals are large, and the phage titers between 70°C~80°C are not measured. Therefore, the phage titer showed abrupt decrease at 80°C in this study. For pH stability experiments, the reason for the phage titer abrupt decrease at pH values 2 and 12 could be similar to that in temperature stability.

Point 5: 2.3. Figure 5: The Figure caption should be shortened and corrected as follows: Figure 5. Phage SPJ41 inhibition of S. typhimurium Sal4 (A), S. enteritidis ATCC 13076 (B) and MDR S. derby A63 (C).

Response 5: Thanks for pointing this out. We have corrected the Figure 5 caption according to your comments, except that “Enteritidis”, “Typhimurium” and “Derby” are written with capital letters and no italics.

Point 6: 2.4. Figure 5: In Figure 5B, the growth of S. enteritidis ATCC 13076 (MOI = 0.01) showed an atypical increase at 2 h of incubation and then decreased at 4 h. Please explain why? The same for S. typhimurium Sal4 (Figure 5A) and MDR S. derby A63 (Figure 5C).

Response 6: Thank you for your comments. The number of bacteria increased during the first 2 h of incubation because low concentrations of phage (MOI = 0.01) could not completely lyse the bacteria. As the phage replicated, the number of phages increased and they were able to lyse more bacteria. So the number of bacteria decreased after 2 h.

Point 7: 2.5. Line 182: “for 6 h, 6 h, 8 h, …..”. I don’t understand, please correct.

Response 7: Thank you for your comments. We have replaced “During phage infection for 6 h, 6 h, 8 h, OD600 values of the three Salmonella strains were always less than 0.2.” with “The bacterial cell densities (OD600) of S. Typhimurium Sal4 and S. Enteritidis ATCC 13076 remained less than 0.2 until 6 h, and until 8 h for MDR S. Derby A63.” in the manuscript.

Point 8: 2.6. Line 183: Replace “the phage treated group with an MOI of 10000” with “the phage-treated group with a MOI of 10000”.

Response 8: Thank you for your comments. We have corrected line 183 according to your comments.

Point 9: 2.7. Figure 6: The Figure caption should be corrected. I didn’t find the notation ns and asterisk (*). Additionally, the meaning of P , 0.01; ***, P , 0.001; ****, P , 0.0001 is unclear.

Response 9: Thank you for your comments. We have replaced “(ns, not significant; *, P , 0.05; **, P , 0.01; ***, P , 0.001; ****, P , 0.0001).” with “Statistical comparisons were performed relative to the control using the two-way ANOVA with multiple comparisons test (**, p < 0.01; ***, p < 0.001; ****, p < 0.0001).” in the manuscript.

Point 10: 2.8. Line 213: Replace “24 hours” with “24 h”.

Response 10: Thanks for pointing this out. We have corrected line 213 according to your comments.

Point 11: 2.9. Figure 7: In Figure 7B, in the treated lettuce samples at 25ºC, the growth of S. enteritidis ATCC 13076 and MDR S. derby A63 showed an increase from 9 and 12 h, respectively until the end of the incubation (24 h). This behavior was not observed in the salmon samples at 25ºC. Please explain why?

Response 11: Thank you for your comments. The treated salmon samples were at 15ºC, not at 25ºC in this study. Phage titers measured at 15ºC and 25ºC were stable in the temperature stability experiments (Figure 1E). In the control group of lettuce samples at 25ºC (Figure 7B), S. Enteritidis ATCC 13076 and MDR S. Derby A63 grew faster than in the control group of salmon samples at 15ºC (Figure 7D). Therefore the inhibitory effect of S. Enteritidis ATCC 13076 and MDR S. Derby A63 was better in salmon treated at 15ºC.

Point 12: 2.10. Figure 7: The bacterial counts in the y-axis in Figures 7A, B, C and D are represented as log10 CFU/Sample. I think that the y-axis should be corrected as log10 CFU/g.

Response 12: Thanks for pointing this out. We have corrected the y-axis of Figure 7 according to your comments.

Point 13: 3. Discussion section: 3.1. Lines 233–238: This information should be moved to the Introduction section since it is not represent a discussion of results.

Response 13: We appreciate the comments. This information is similar to that expressed in lines 60-63 of the last paragraph of the Introduction. We have deleted the first two sentences of this information and put the last sentence at the beginning of the next paragraph.

Point 14: 4. Materials and Methods section: 4.1. Lines 363–370: Why the pH stability of phage SPJ41 was not conducted using buffers instead of NaOH or HCl?

Response 14: Thank you for your comments. For pH stability in this experiment, SM was adjusted to pH 2–13 with NaOH or HCl. We have replaced “900 μL fresh TSB pre-adjusted from pH 2 to 13 with NaOH or HCl.” with “900 μL of SM buffer at different pH values (pH 2-13, adjusted using NaOH or HCl).”in the line 365–366.

Point 15: 4.2. Lines 363–370: Why different volumes (100 µL or 200 µL) and phage suspensions (109 PFU/mL or 108 PFU/mL) were used to evaluate the pH and temperature stability?

Response 15: Thank you for your comments. 100 μL of phage suspension was used in the temperature stability experiment, which we have corrected in the manuscript.

100 µL of phage suspension (109 PFU/mL) was used for pH stability as it was then added to 900 µL of SM buffer at different pH to a final concentration of 108 PFU/mL. Whereas 108 PFU/mL phage suspension was used directly in the temperature stability experiment, therefore, the final concentration of phage was the same for the temperature and pH stability.

Point 16: 4.3. Lines 363–370: Why didn´t the authors perform the evaluation of the pH and temperature stability using the response surface methodology? With this approach, the true optimum values for pH and T could be obtained and the number of experiments could be considerably reduced.

Response 16: Thank you for your comments. Response surface methodology can use less experiments to establish a more accurate mathematical model and find the best combination of experimental variables and the optimal response value. In the study of phage stability, the purpose of this experiment is to understand the scope of application of the phage as an antibacterial agent in the food industry and whether it can be used in harsh environments, rather than to obtain the optimal values of pH and T.

Point 17: 4.4. Lines 391–399: Why didn´t the authors calculate the lethal dose (LD50) of phage SPJ41 against S. typhimurium Sal4, S. enteritidis ATCC 13076 and MDR S. derby A63? With the data measured in this assay (OD600 values), the corresponding inhibition curves and LD50 of phage SPJ41 for each strain could be obtained.

Response 17: Thank you for your comments. The median lethal dose (LD50) refers to the minimum number of bacteria or toxin required to kill half of a certain animal of a certain weight or age through a specified infection route within a specified period of time. This study does not involve these experiments, so LD50 was not calculated.

Point 18: 4.5. Line 403: Replace “CFU/ml” by “CFU/mL”.

Response 18: Thanks for pointing this out. We have corrected line 403 according to your comments.

Point 19: 4.6. Line 425: Replace “24 hours” by “24 h”.

Response 19: Thanks for pointing this out. We have corrected line 425 according to your comments.

Point 20: 4.7. Line 438: Please provided the year of the GraphPad Prism 9.0 program used.

Response 20: Thank you for your comments. The GraphPad Prism 9.0 program is dated October 22, 2020.

Point 21: 4.8. Lines 438–439: Replace “A p value of < 0.05 was considered as statistically significant” by “Differences were considered statistically significant for p < 0.05”.

Response 21: Thank you for your comments. We have corrected line 438–439 according to your comments.

References

  1. Yi, Y.; Abdelhamid, A. G.; Xu, Y.; Yousef, A. E. Characterization of broad-host lytic Salmonella phages isolated from livestock farms and application against Salmonella Enteritidis in liquid whole egg [J]. LWT, 2021, 144: 111269.
  2. Park, H.; Kim, J.; Kim, H.; Cho, E.; Park, H.; Jeon, B.; Ryu, S. Characterization of the lytic phage MSP1 for the inhibition of multidrug-resistant Salmonella enterica serovars Thompson and its biofilm [J]. International journal of food microbiology, 2023, 385: 110010.

Reviewer 2 Report

I have no particular comments to make on the article submitted to me. The whole article was developed according to the criteria of scientific research and the results obtained seem very interesting to me. I have marked some of my observations directly on my review file, which I enclose.
My compliments to the authors.

Author Response

Response to Reviewer 2 Comments

Dear Reviewer:

Thank you for your reply concerning our manuscript entitled “Characterization of phage vB_SalM_SPJ41 and reduction of risk of antibiotic-resistant Salmonella enterica contamination in two ready-to-eat foods” (ID: antibiotics 2174202). Those comments are all valuable and helpful for improving our paper. We have revised the manuscript according to the comments and highlighted the revised portion (red font is the modified content). The main correction and the respond to reviewers’ comments are as follows:

Point 1: I have no particular comments to make on the article submitted to me. The whole article was developed according to the criteria of scientific research and the results obtained seem very interesting to me. I have marked some of my observations directly on my review file, which I enclose.

Response 1: Thanks for the very positive comments on our work. We have carefully corrected the parts highlighted in yellow in the manuscript according to your comments.

Point 2: Line 298: Here, as elsewhere in the article, the authors reported et al without the period. instead in other points of the article et al. is written with a period. univormate with Dot et al.

Response 2: Thanks for pointing this out. We have corrected all "the authors reported et al without the period." in the manuscript to "et al. is written with a period.".

Point 3: Line 306: Why is the Materials and Methods chapter after the Discussion and not before?

Response 3: Thank you for your comments. Since the Materials and Methods chapter of the “Antibiotics” manuscript template is after the Discussion, we also place it later.

Reviewer 3 Report

·       This is a very interesting paper and generally well done. Nevertheless, a few modifications and improvements are needed before it may be considered for publication. Authors The name of phage may be simplified

·       Myoviridae is a family of phages characterized by their morphology (coincident with the one of your phage) but essentially because the contractile tail that may be observed in your electron micrographies. This should be highlighted.

·        The “one-step growth curve of phage SPJ41” is not presented since Fig. 1c is not a one-step growth curve probably the experiments was not en ough well done”

·       “126 PFU/cell”  should be “126 PFU/infected cell” Nevertheless the one-step growth curve should be repeated

·       In section 2.7 it may have interest to consider the effective inhibition for the first 5 hours.   

·       Some sentences need some improvement e.g.

P2: Bacteriophage is a virus…..

Should be

A bacteriophage is a virus…..

Or better

Bacteriophages are  virus…

P2 contamination------contaminants

Author Response

Response to Reviewer 3 Comments

Dear Reviewer:

Thank you for your reply concerning our manuscript entitled “Characterization of phage vB_SalM_SPJ41 and reduction of risk of antibiotic-resistant Salmonella enterica contamination in two ready-to-eat foods” (ID: antibiotics 2174202). Those comments are all valuable and helpful for improving our paper. We have revised the manuscript according to the comments and highlighted the revised portion (red font is the modified content). The main correction and the respond to reviewers’ comments are as follows:

Point 1: This is a very interesting paper and generally well done. Nevertheless, a few modifications and improvements are needed before it may be considered for publication. Authors The name of phage may be simplified.

Response 1: Thanks for the very positive comments on our work. We referred to the paper of Kropinski et al. [1] to name the phage, and the guidelines are as follows.

(a) The name is preceded by the prefix vB (bacterial virus).

(b) The host abbreviation system used for restriction enzymes based upon the genus and species e.g. Eco (Escherichia coli).

(c) The name ends with the single letter virus family e.g. M (Myoviridae).

(d) The specific laboratory designation or common name remains the responsibility of the researchers.

The name "SPJ41" is based on the initial letter “S” and number “4” of the host bacteria Salmonella Sal4, the initial letter “P” of the phage and the initial letter “J” of the Chinese spelling of chicken manure. So we finally named the phage “vB_SalM_SPJ41” and the abbreviated name “SPJ41”.

Point 2: Myoviridae is a family of phages characterized by their morphology (coincident with the one of your phage) but essentially because the contractile tail that may be observed in your electron micrographies. This should be highlighted.

Response 2: Thank you for your comments. We have highlighted phage tail contractile in TEM according to your comments, and we have replaced "long tail" with "long contractile tail" in line 74.

Point 3: The “one-step growth curve of phage SPJ41” is not presented since Fig. 1c is not a one-step growth curve probably the experiments was not enough well done”

Response 3: Thank you for your comments. We have tried 3 times and the results were the same.

Most phages follow a one-step growth curve pattern. The Salmonella phages (OSY-STA and OSY- SHC) reported by Yi et al. is similar to the phage vB_SalM_SPJ41 in replication pattern [2]. Yi et al. called it “replication kinetic” instead of “one-step growth curve”. The burst size of phage OSY-STA and OSY-SHC was only calculated for the first stage. So, we have corrected the “one-step growth curve” to “replication kinetic” in the Results section: 2.3, Materials and methods: 4.6 and Discussion section. We emphasized in the Results section: 2.3 that the growth curves are in two phases, and the first phase burst size was calculated. We have also modified the calculation of burst size in the Methods section: 4.6 as “Relative burst size = (final phage titer – initial phage titer) / initial phage titer”.

Point 4: “126 PFU/cell” should be “126 PFU/infected cell” Nevertheless the one-step growth curve should be repeated.

Response 4: Thank you for your comments. We have repeated 3 times and the results were the same. We recalculated the size of the first phase as 255 PFU/cell. We have revised “126 PFU/cell” to “255 PFU/infected cell”.

Point 5: In section 2.7 it may have interest to consider the effective inhibition for the first 5 hours.   

Response 5: Thank you for your comments. We added “In the phage treatment group except the MOI of 0.01, the measured OD600 value was always lower than 0.1 in the first 5 hours.” in section 2.7.

Point 6: Some sentences need some improvement e.g.

P2: Bacteriophage is a virus…..

Should be

A bacteriophage is a virus…..

Or better

Bacteriophages are virus…

P2 contamination------contaminants

Response 6: Thanks for pointing this out. We have corrected these sentences according to your comments.

References

  1. Kropinski, A. M.; Prangishvili, D.; Lavigne, R. Position paper: The creation of a rational scheme for the nomenclature of viruses of Bacteria and Archaea [J]. Environmental Microbiology, 2009, 11(11): 2775-2777.
  2. Yi, Y.; Abdelhamid, A. G.; Xu, Y.; Yousef, A. E. Characterization of broad-host lytic Salmonella phages isolated from livestock farms and application against Salmonella Enteritidis in liquid whole egg [J]. LWT, 2021, 144: 111269.

Round 2

Reviewer 3 Report

Thank you for addressing most of my previous comments. Nevertheless I dont think the so called "one step growth curve" may be published as it is . The authors wrote that they did three times the experiment and they wrote the name was changed, although in the footnote of the figure still appears as “one step growth” and the graph is identiocal to the previous one.  May the authors provide a graph average of the three replicates with  deviations?. The reference provided (Yi et al., ) is not available for me; may the authors inform the journal in which it was published. Nevertheless the same authors have published an announcement (Draft Genome Sequence of the Lytic Salmonella Phage OSY-STA, Which Infects Multiple Salmonella Serovars Microbios announcements ASM) but here no reference of the multiplication kinetics may be found. I firmly believe that such a graph should not be published unless more sound basis  or some available reference may be provided.

Round 3

Reviewer 3 Report

As far as I realize you want to puilish your replication curve as it is, go ahead.